# Host relatedness and landscape connectivity shape pathogen spread in the puma, a large secretive carnivore

Nicholas M. Fountain-Jones [1,2 ✉], Simona Kraberger[3], Roderick B. Gagne [3], Daryl R. Trumbo[4], Patricia E. Salerno[4,5], W. Chris Funk [4], Kevin Crooks[6], Roman Biek [7], Mathew Alldredge[8], Ken Logan[9], Guy Baele[10], Simon Dellicour [10,11], Holly B. Ernest[12], Sue VandeWoude [3], Scott Carver [2] & Meggan E. Craft [1]

Urban expansion can fundamentally alter wildlife movement and gene flow, but how urbanization alters pathogen spread is poorly understood. Here, we combine high resolution host and viral genomic data with landscape variables to examine the context of viral spread in puma (*Puma concolor*) from two contrasting regions: one bounded by the wildland urban interface (WUI) and one unbounded with minimal anthropogenic development (UB). We found landscape variables and host gene flow explained significant amounts of variation of feline immunodeficiency virus (FIV) spread in the WUI, but not in the unbounded region. The most important predictors of viral spread also differed; host spatial proximity, host relatedness, and mountain ranges played a role in FIV spread in the WUI, whereas roads might have facilitated viral spread in the unbounded region. Our research demonstrates how anthropogenic landscapes can alter pathogen spread, providing a more nuanced understanding of host-pathogen relationships to inform disease ecology in free-ranging species.

[1] Department of Veterinary Population Medicine, University of Minnesota, St Paul, MN 55108, USA. [2] School of Natural Sciences, University of Tasmania, Hobart, Australia 7001. [3] Department of Microbiology, Immunology, and Pathology, Colorado State University, Fort Collins, CO 80523, USA. [4] Department of Biology, Graduate Degree Program in Ecology, Colorado State University, Fort Collins, CO 80523, USA. [5] Universidad Regional Amazónica IKIAM, Km 7 vía Muyuna, Tena, Ecuador. [6] Department of Fish, Wildlife, and Conservation Biology, Colorado State University, Fort Collins, CO 80523, USA. [7] Institute of Biodiversity, Animal Health and Comparative Medicine, University of Glasgow, Glasgow G12 8QQ, UK. [8] Colorado Parks and Wildlife, Fort Collins, CO 80526, USA. [9] Colorado Parks and Wildlife, Montrose, CO 81401, USA. [10] Department of Microbiology, Immunology and Transplantation, Rega Institute, KU Leuven, Herestraat 49, 3000 Leuven, Belgium. [11] Spatial Epidemiology Lab (SpELL), Université Libre de Bruxelles, CP160/12 50, av. FD Roosevelt, 1050 Bruxelles, Belgium. [12] Wildlife Genomics and Disease Ecology Lab, Department of Veterinary Sciences, University of Wyoming, Laramie, WY 82070, USA. ✉email: Nick.FountainJones@utas.edu.au

Understanding how pathogens spread through populations remains a fundamental challenge. The extent to which pathogen spread reflects movement patterns of their hosts is enigmatic but important for controlling disease[1,2]. If pathogen spread mirrors host gene flow, host genetic structure/differentiation could be a valuable proxy for pathogen spread and be used as a basis to inform disease control[3–7] (e.g., male vampire bat [*Desmodus rotundus*] genetics closely mirrors phylogenetic structure of rabies[8]). A close relationship between host gene flow and pathogen spread may also be evidence for increased transmission between related conspecifics, and could affect evolutionary pressures on the pathogen, as closely related hosts may be more likely to have similar immune environments[9]. In contrast, if host gene flow and pathogen spread are decoupled, fine-scale patterns of host movement, for example, may best predict spread and thus inform the strategy employed for disease control[2,10,11]. The patterns of pathogen spread can also be influenced by characteristics of the pathogen itself, where host-specific and directly transmitted pathogens likely have the greatest concordance with host gene flow, relative to multi-host and environmentally transmitted pathogens. Given this range of scenarios, a better understanding of how host relatedness and environmental predictors drive these processes would improve our estimates of pathogen spread in heterogeneous landscapes.

Urbanization is one of the most destructive and large-scale of all anthropogenic landscape fragmentation processes, but how urbanization shapes pathogen spread in particular is still not well understood[12,13]. As urban development fragments habitats and introduces barriers (the wildland–urban interface, WUI), it can cause reduced host gene flow between populations[10,14], altered animal behaviour (for example, animals becoming more nocturnal to avoid humans[15,16]), and changes in feeding[17] and movement[18] patterns. If these anthropogenic impacts on host behaviour affect transmission dynamics, they may manifest in the demographics of pathogen populations[19] (e.g., if transmission events are happening rapidly, the pathogen's effective population size may be exponentially increasing[19]). Comparing the factors that shape pathogen spread in populations that are affected by urbanization is often difficult due to a lack of high-resolution data (i.e., coupled host and pathogen genomic data for most individuals) or comparable populations (i.e., well-sampled populations impeded and unimpeded by urbanization). Quantifying how urbanization can affect host gene flow, and how this in turn impacts the transmission dynamics and spread of pathogens, can help address this important research gap.

Here, we determine how landscape variables (including those associated with urbanization) and host relatedness affect pathogen spread and transmission in puma (*Puma concolor*). Puma are useful indicators of the effects of urbanization on wildlife as they are sensitive to urban development[17,20] but can persist in areas impacted by urbanization provided sufficient landscape connectivity (e.g., ref. [21]). As puma foraging, movement, and other behaviours are altered by urban development[17,22,23], this species offers a valuable case study for how pathogen spread can be effected by urbanization—subject matter that is increasingly important for wildlife conservation and management[24,25]. We utilize data we collected from 217 pumas sampled from two geographically distinct regions (~500 km apart): one region bounded by the wildland–urban interface (hereafter the WUI) and the other in a more wild and rural setting relatively unbounded by anthropogenic development (hereafter UB). Our previous work found limited gene flow between pumas from these two regions but similar levels of genetic diversity[26]. From individuals in both regions, we collated high-resolution host genomic and spatial data alongside puma feline immunodeficiency virus (FIV$_{pco}$) sampled from the same individuals. FIV$_{pco}$ is a rapidly

evolving retrovirus[27] endemic to puma populations, and is thought to be predominantly transmitted horizontally via aggressive encounters[28], although vertical transmission has been documented by phylogenetic analyses[29]. Because FIV$_{pco}$ is essentially apathogenic in puma[30,31], it is an ideal model pathogen to understand transmission dynamics in wild systems without potential confounding effects of disease on behaviour and demography[29,32]. We examine what factors impact FIV$_{pco}$ spread using a novel pipeline synthesizing phylodynamic, phylogeographic and landscape genetic techniques (an ecophylogenetic approach[33]). We employ this pipeline to test for (1) differences in FIV$_{pco}$ demographic histories and transmission dynamics across regions, (2) concordant patterns of host relatedness, viral phylogenetics and spatial distance, and (3) the relative roles of host relatedness and landscape predictors, such as urban development, in shaping the pattern of spread of the virus. We hypothesized that as anthropogenic factors impact puma movement (e.g., ref. [34]) and gene flow[26] at the WUI, that transmission opportunities would be restricted and spatial proximity and host relatedness would be more important in shaping spread in this region.

## Results

Of the 217 individuals we tested, we found that FIV$_{pco}$ prevalence was higher in the WUI than UB puma (58% vs 41%, $p = 0.04$ (two-sample test for equality of proportions), see Table S1 for site and population characteristics). We sequenced FIV$_{pco}$ from a total of 46 animals representing most of the infected animals in both populations over a 10-year period. For 43 of the pumas, we obtained both FIV$_{pco}$ sequences (the conserved *pol, ORFA* and *env* genes representing 36% of the FIV genome) and the corresponding puma genomic data (consisting of a dataset of 12,444 neutral single nucleotide polymorphisms [SNPs] per individual[26]).

**Region-specific viral demographic histories**. We found not only distinct demographic histories in the viruses circulating in the WUI and UB regions, but also differing FIV$_{pco}$ subtypes. Bayesian time-scaled phylogenetic analysis of the FIV$_{pco}$ sequences revealed two co-circulating FIV$_{pco}$ subtypes: FIV$_{pco}$ CO, circulating among pumas in both regions (Fig. 1a) and FIV$_{pco}$ WY, which was only detected in the UB after having been previously detected in puma in Wyoming (Fig. 1b; see Fig. S1 for a maximum-likelihood tree that illustrates the broader phylogenetic context of these two subtypes across North America). Within FIV$_{pco}$ CO, we identified three clades (I, II and III, Fig. 1a) that had contrasting and landscape-specific demographic histories (Fig. S2). Clade I had been circulating predominantly in the WUI since ~1995 (95% high posterior density interval (HPD): 1984–2003), and the effective population size of this clade has been gradually increasing through time (Fig. S2). Clade II showed a similar trajectory in population size (Fig. S2) and was found in puma from both regions, which is potentially indicative of long-distance dispersal of FIV$_{pco}$ (Fig. 1a). Clade III, in contrast, predominantly circulated in the UB, but had a much more distinctive demographic pattern (Fig. S2). We estimated that Clade III began circulating in the WUI in 2001 (95% HPD: 1992–2006) and arrived in the UB in 2006 (95% HPD: 2003–2008), afterwards going through a period of population growth which plateaued around 2012 (Fig. S3). In contrast, we found that FIV$_{pco}$ WY has likely circulated at low prevalence in the UB for over a hundred years (Fig. S3) and had a slower estimated evolutionary rate than FIV$_{pco}$ CO (Table S2).

**Divergent patterns of viral and host relatedness across regions.** Overall, despite regional fidelity, the FIV$_{pco}$ phylogeny did not

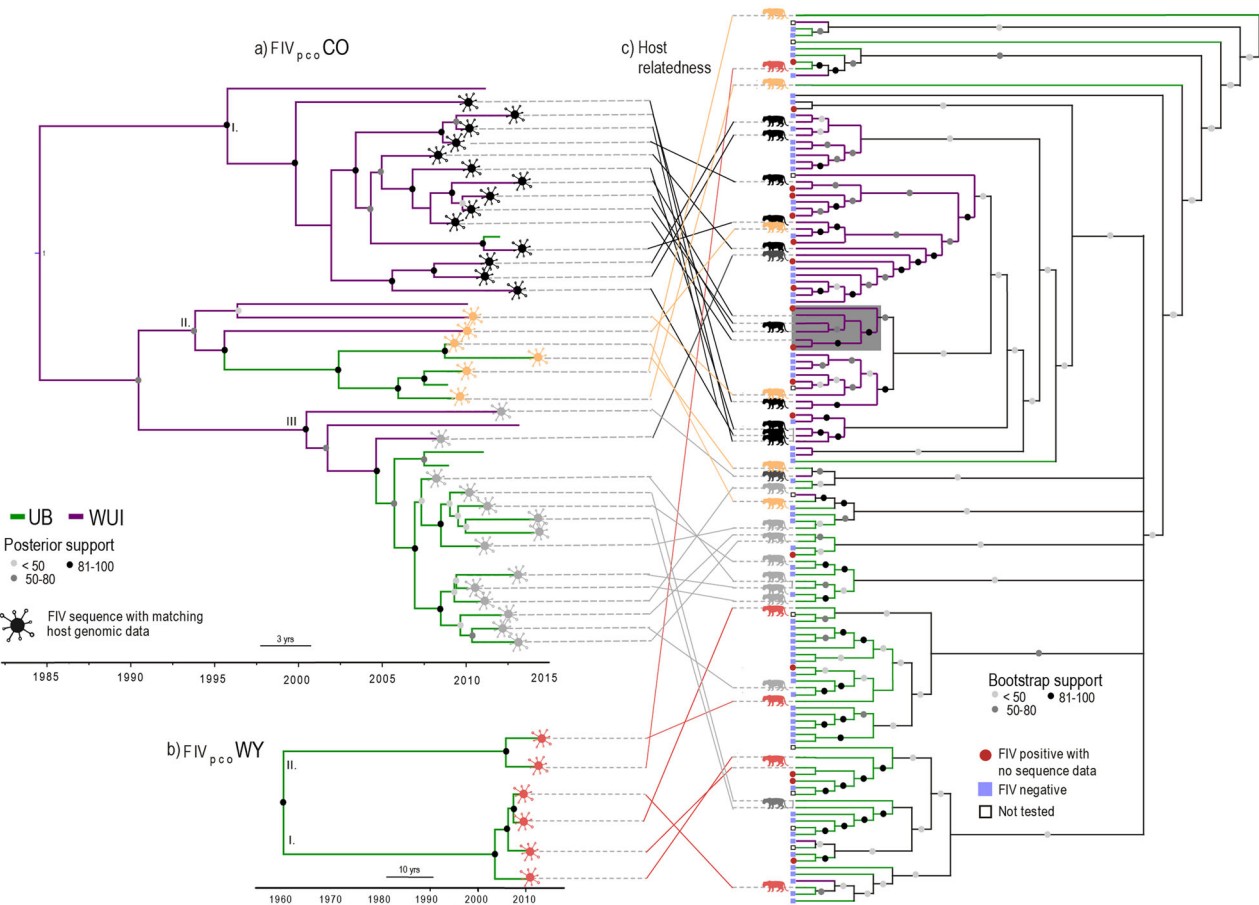

**Fig. 1 Tanglegram revealing how FIV$_{pco}$ phylogenetic relationships overall mapped imprecisely onto the puma relatedness cladogram in our study regions. a** Bayesian time-scaled phylogenetic tree for FIV$_{pco}$ subtype CO found in both the wildland–urban interface (WUI) and unbounded (UB) regions. **b** Bayesian time-scaled phylogenetic tree for FIV$_{pco}$ subtype WY which was only found in the UB. I–III represent the different clades identified using tree structure analysis[69]. Virus branch colours are based on population assignment posterior values from our FIV$_{pco}$ subtype CO discrete trait analysis. **c** Host relatedness cladogram constructed using singular value decomposition (SVD) quartets[37] based on over 12,000 SNPs from 130 individual puma across both study areas. The grey shaded box encompasses related individuals with phylogenetically similar FIV$_{pco}$ isolates. Virus symbols (from panels **a** and **b**) are coloured based on viral lineage membership. This colour matches the puma infected with that isolate (puma silhouettes **c**) and the lines connecting each isolate to each host in the tanglegram. Tips without virus symbols indicate that there was no matching host genomic data for this FIV$_{pco}$ isolate. Branch colours indicate which region each individual puma and matching virus was sampled from (WUI or UB).

map closely onto the puma relatedness cladogram; yet there was some localized evidence for concordance between the two in the WUI region (grey boxes, Fig. 1a–c). For example, four related individuals in the WUI were infected with phylogenetically similar FIV$_{pco}$ (dark grey box, Fig. 1a–c) and were also captured in close spatial proximity to each other (dashed circle, Fig. 2d). In contrast, there was limited evidence of similar patterns in the UB as the most phylogenetically similar FIV$_{pco}$ isolates were sampled across unrelated individuals. In the UB, a significantly higher proportion of individuals in each 'neighbourhood' (i.e., pumas likely to have home-range overlap) were qPCR negative for FIV$_{pco}$ than puma in the WUI (Mann–Whitney U Test, $p = 0.007$, Fig. S4). In addition, in both regions, there was qualitative evidence for spatial structuring with some FIV$_{pco}$ lineages being locally dominant (e.g., CO clade I and III in Fig. 2). However, overall FIV$_{pco}$ spread was a complex mixture of local and longer-distance jumps across both landscapes with uninfected individuals captured at the same time and within 500 m of infected individuals in both regions (Fig. 2). Similarly, individuals captured within months of each other at the same location were commonly infected with phylogenetically distinct FIV$_{pco}$ subtypes or clades (Fig. 2).

**Predictors of FIV spread are region-dependent**. We employed generalized dissimilarity models[35] (GDM) and maximum likelihood of population-effects[36] (MLPE) to test how host and landscape shaped two components (the overall phylogeographic pattern and lineage dispersal velocity) of spread in each region. Landscape variables for both techniques were formulated using a resistance/conductance approach[37]. We calibrated our resistance/conductance costs based on expert opinion as well as optimizing the landscape variables using host genetic distance using the Resistance GA routine[35]. In the UB, our optimization approach revealed that none of the landscape variables (Table S3) explained host gene flow more than the null model (i.e. models were >2 AICc units higher than the null or the model with no landscape variables included, Table S4). Subsequently, no host-genetic optimized surfaces were included in the UB model (see 'Methods'). In contrast, our optimisation approach identified a stronger impact of spatial proximity on host gene flow in the WUI (Table S5). Canopy cover and urban land cover univariate models were also within 2 AICc units of the spatial proximity model, thus we combined these variables together to generate a multivariate, host genetics optimised resistance surface (hereafter called host-optimized resistance

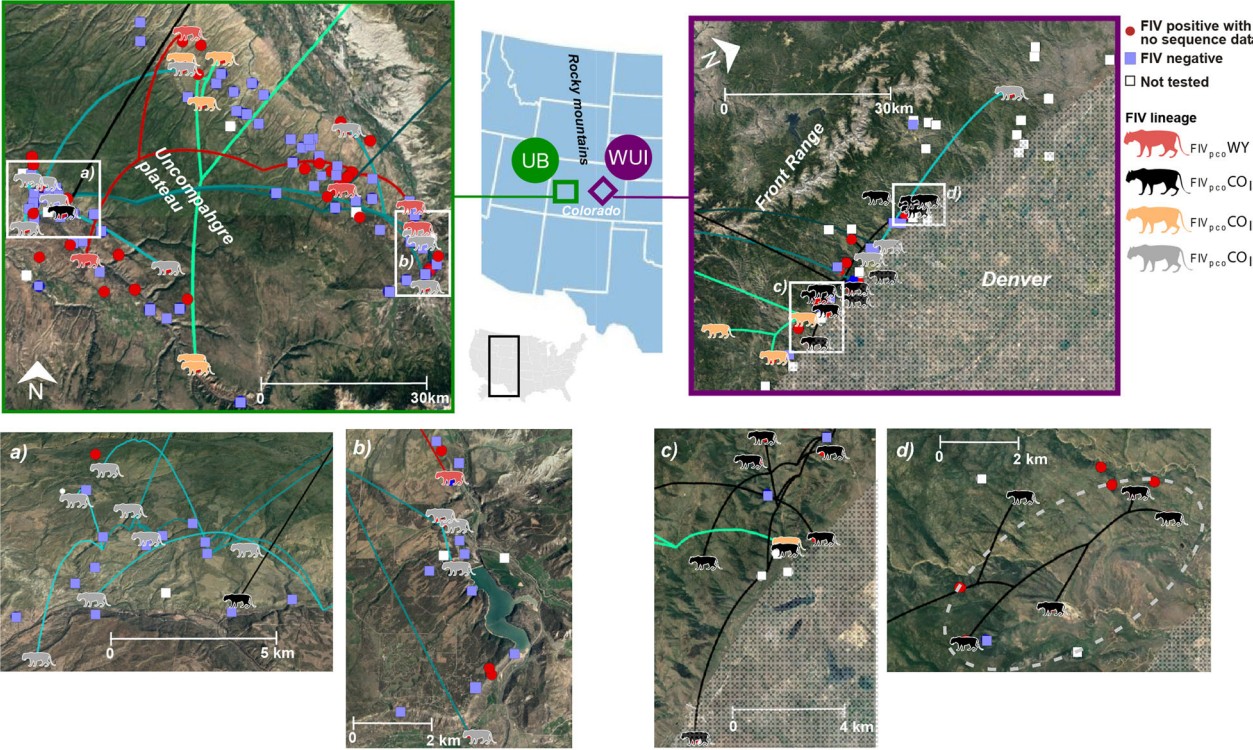

**Fig. 2 Spatially projected FIV phylogenies showing the configuration of FIV$_{pco}$ spread in the unbounded (UB) and wildland–urban interface (WUI) regions.** Panels **a–d** represent enlarged sections of each area. Tree tips represent the capture locations. Lines indicate FIV$_{pco}$ estimated branch locations, with colours of each line as well as the coloured puma symbols reflecting FIV$_{pco}$ subtype or lineage membership based on tree structure analysis[69]. The dashed circle indicates the group of individuals in the WUI in which the viral phylogeny mapped onto the host cladogram (grey boxes in Fig. 1). Grey shading shows the extent of the Denver metropolitan areas. Branches coming from outside the study area in the top panels are the branches connecting each region. See Supplementary Data 1–4 for .kml files to recreate this map. FIV$_{pco}$ or host genomic data could not be tested in some individuals (white boxes).

surface). The host-optimized resistance surface was strongly correlated with interpolated host genetic resistance (Mantel $\rho = 0.76$, $p = 0.001$). We thus present models in the WUI region with host genetic resistance and host-optimised resistance included separately. In both regions, to tease apart the effect of host genetics and landscape on FIV spread, we also included the non-host genetics optimized landscape resistance/conductance surfaces (hereafter called landscape variables) in our models as well as spatial proximity and host variables.

Strikingly, landscape and host variables explained significant variation of FIV$_{pco}$ spread for the WUI only. In the WUI, our GDM models explained 20% of total model deviance ($p = 0.012$) but only 7% in the unbounded population ($p = 0.23$). Moreover, the most important variables that shaped spread in each case were different (Fig. 3a). To support these results, we compared our non-linear GDM models to MLPE. One advantage of the GDM method over MLPE and other methods is that it can capture non-linear associations between response and predictor matrices. However, model performance of MLPE has been more rigorously evaluated compared to GDM on landscape genetic datasets[38]. Here, we highlight factors that explain the most deviance in our GDM models and are within two log units of the best performing MLPE models using BIC (Tables S6 and S7). In the UB, FIV$_{pco}$ spread was associated with roads (i.e., individuals more connected by roads had similar FIV isolates, Fig. 3b, Table S6). In contrast, the viral spread in the WUI was shaped by spatial proximity coupled with host relatedness and impervious surface (Fig. 3a, Table S7). As spatial proximity decreased, so did the FIV$_{pco}$ patristic distance between individuals; neighbouring individuals

shared more phylogenetically similar FIV$_{pco}$ isolates (Fig. 3c). Similarly, as host relatedness decreased, so did FIV$_{pco}$ patristic distance (i.e., related individuals were more likely to share phylogenetically similar FIV$_{pco}$ isolates, Fig. 3e). A different measure of individual genetic distance (the Smouse measure[39]) did not alter our results. We found a similar positive relationship between FIV$_{pco}$ patristic distance and impervious surface resistance (Fig. 3e). Furthermore, in complement to our relatedness measure, we also included host genetic resistance in our GDM models (see 'Methods' for details). Individuals in the WUI with low host genetic resistance values had more similar phylogenetically FIV$_{pco}$ isolates (Fig. 3f). However, this relationship plateaued with dissimilarity values of over 0.05 and was not significant ($p = 0.38$).

We also found that there were region-specific impacts of landscape on FIV$_{pco}$ lineage dispersal velocity. Our analysis revealed that elevation tended to act as a conductance factor increasing the dispersal velocity of FIV$_{pco}$ lineages in the WUI, whereas none of the predictors we measured had any substantial effect on lineage dispersal velocity in the UB (positive $Q$ distribution and associated Bayes factor support >3[40]; see Fig. S6 and Table S7 for a list of tested landscape factors). Furthermore, warmer minimum temperatures (as measured in the coolest month) tended to act as a resistance factor decreasing lineage dispersal velocity in the same region. Taken together, these results indicate that in the WUI, FIV$_{pco}$ tended to spread faster through colder, higher elevation areas less suitable for puma habitat. In the WUI, the areas of higher elevation tended to be away from the urban edge.

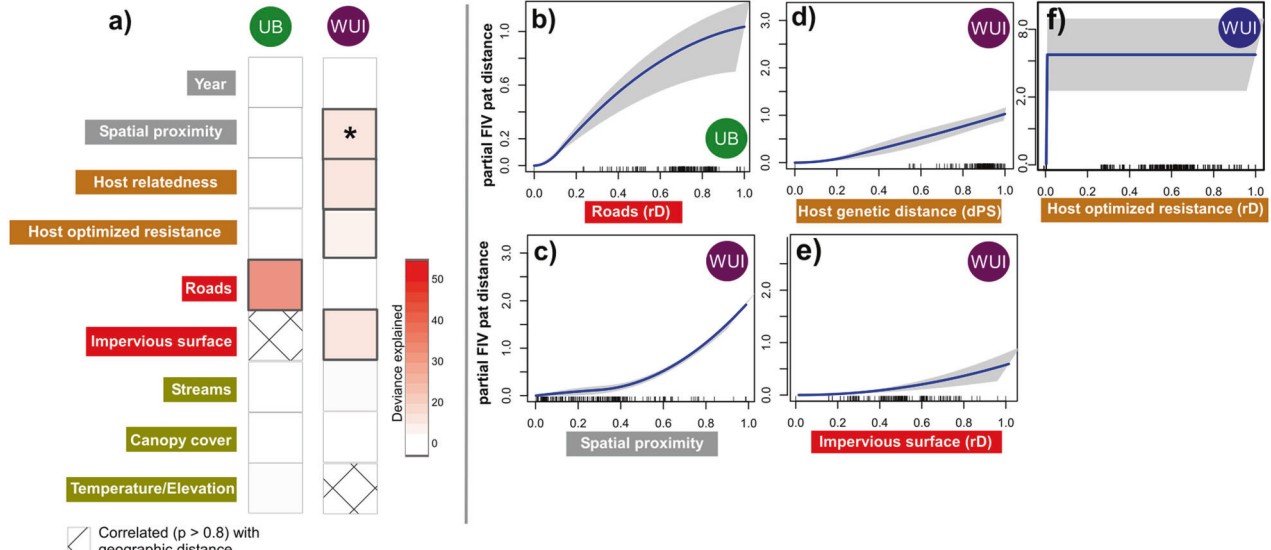

**Fig. 3 Results from the GDM analyses of FIV$_{pco}$ spread. a** Heat map showing the deviance explained by each predictor in the unbounded (UB) and wildland–urban interface (WUI) *$p$ value 0.01–0.05. Predictor labels in grey boxes = spatiotemporal predictors, predictors in orange = host genetic measures, predictors in red = associated with anthropogenic development, predictors in olive green = natural landscape features. The right panel shows partial effect plots (all other variables held at the mean) of the GDM-fitted I-splines showing the relationship between partial FIV$_{pco}$ patristic distance for the variables selected in the UB final model (**b**, see 'Methods') and WUI (**c–f**) in order of importance. Pat distance = patristic distance. Lines across the $x$-axis represent a rug plot showing the distribution of the data and the grey surrounding the blue line (the fit) indicates 95% confidence intervals. rD: Resistance distance. dps: Proportion of shared alleles. See Tables S6/S7 for corresponding MLPE results. See Fig. S5 for the model with host genetic resistance is substituted for optimized host resistance.

## Discussion

Our multifaceted approach linking landscape, host genomic, and pathogen genomic data uncovered unique landscape-specific relationships with FIV$_{pco}$ spread, indicative of altered epidemiological dynamics associated with urban landscape structure. We found that spatial proximity was positively associated with FIV$_{pco}$ spread, but only in the WUI (Wildland Urban Interface). In the WUI, host relatedness, host-optimised resistance and impervious surface also had a minor association with FIV$_{pco}$ spread. These landscape factors mirrored the factors shaping host gene flow in the WUI[26], providing support that host movement and viral spread were more intimately related in the WUI than in the UB (Unbounded) region. However, while there was some evidence for concordance between the FIV$_{pco}$ phylogeny and host cladogram in the WUI, they did not map precisely onto each other in either region (Fig. 1), indicating that transmission occurs outside of related individuals. There was little evidence of FIV$_{pco}$/host concordance in the UB where host relatedness did not shape phylogeography and entirely different sets of predictors shaped host gene flow (spatial proximity and tree cover[26]). These opposing patterns between host gene flow and viral spread could reflect regional differences in transmission. One potential scenario is that transmission between neighbouring related conspecifics may be more likely in the WUI due to altered puma movement, dispersal patterns[22,23] and foraging behaviours[17]. The urban development that impacts the WUI is linear (i.e., where the Great Plains meet the Front Range of the Rocky Mountains). This linear development restricts juvenile dispersal[23] and could lead to more opportunities for transmission among related conspecifics (i.e., individuals establish home ranges close to their parents). Our previous work supports this hypothesis as we demonstrated that family units were more clustered in space, where puma genetic distance per kilometre and sub-structure was greater in the WUI compared to the UB[26]. There has not been an extensive evaluation of relatedness in neighbouring home-range females, but female matrilines (groups of maternally-related females) are

known to occur[41,42]. Furthermore, we found evidence that infection was more clustered in space in the WUI compared to the unbounded one, supporting the idea that spatial proximity increases transmission risk between related individuals in the WUI. This shift in transmission risk may reduce evolutionary pressure on the virus due to factors such as similarity in immune profile[9].

Roads, common but mostly unpaved in this UB region, were a modest predictor of spread in the UB. Radiotelemetry has shown that puma often move using unpaved roads[43] and rapid viral evolution potentially allowed us to detect this modest effect. There was a smaller impact of roads on host gene flow[26] which supports the idea that FIV phylogenetics may capture more contemporary movement patterns impossible to detect using host genetics alone[10,11,44]. When host gene flow and viral spread are decoupled, as was the case in the UB, the rapid accumulation of viral mutations may conversely obscure historical trends in connectivity[45]. This could explain why we detected no effect of tree cover on FIV$_{pco}$ spread even though puma are known to have a preference for tree cover to disperse and hunt[46,47]. The altered epidemiological history of the clade, dominant in the UB compared to all other detected clades, may reflect or be a consequence of relatively unrestricted spread in the UB (Fig. S3, FIV$_{pco}$ CO clade I). We postulate that recent arrival of FIV$_{pco}$ CO clade I in the UB and signature of rapid expansion[19] across the Uncompahgre Plateau (Fig. S2) may only have been possible in a region where viral spread itself was unbounded. Further work is needed to assess the temporal dynamics of FIV$_{pco}$ in the UB. The high elevation Uncompahgre Plateau (averaging 2900 m a.s.l, Fig. 2) did not shape viral spread, even though puma are known to have a preference for not dispersing across cold, high altitude divides[46]. In contrast, we found that higher altitude areas increased dispersal velocity in the WUI. As most human activity in the Front Range (Fig. 2) occurs in lower altitude areas, it is plausible that increased viral velocity in higher altitude areas is a product of the host's avoidance of the human 'super predator'[17,48].

FIV transmission events have been shown to be more likely to occur further from the urban edge in bobcat populations[26] and the same may apply for pumas in the WUI region here. Velocity may also be faster through higher elevations in the Front Range as puma are more likely to rapidly move through unsuitable habitat. Unsuitable habitat has also been demonstrated to increase the velocity of rabies lineages in dogs[49]. We acknowledge that in this study we did not sample all pumas in the system nor between the two sampled regions, and that $FIV_{pco}$ could not always be sequenced. This could mean that we missed, for example, some $FIV_{pco}$ lineages that could have altered our inference about $FIV_{pco}$ patterns. Nonetheless, this did not compromise our ability to gain complementary insights into the drivers of host connectivity by combining high-resolution host and pathogen genomic data, which would have been impossible to detect with either host or pathogen data alone.

Our findings have pathogen and host management implications, as we demonstrated that spatial proximity and host relatedness may be relevant predictors of pathogen spread in regions impacted by urban development. This may mean that the difficult task of disease control in a large apex predator (such as vaccinating against feline leukaemia virus in a puma population[25]) may be more tractable in bounded populations. Because juveniles set up home ranges near their parents, dispersal events important for pathogen jumps in the landscape are more constrained. Further, as host gene flow and viral spread were tightly linked in the WUI, targeting individuals where gene flow was less constrained by impervious surface (further from the urban areas) may also reduce spread. Our work provides a valuable case study of how landscape context and host relatedness can be important in disease management plans. As urban landscapes continue to expand, improving our understanding of how heterogeneous landscapes and host relatedness alter pathogen transmission will be increasingly important.

## Methods

**Samples**. Puma blood and tissue samples were collected from 103 individuals (48 males, 55 females) between 2005 and 2014 in the UB and 110 individuals (43 males, 54 females, 12 undetermined) between 2003 and 2015 from the WUI as part of monitoring efforts by Colorado Parks and Wildlife in the Rocky Mountain Range of Colorado, USA[23,50]. This sampling effort is likely to represent a large proportion of the resident puma present in both regions during the sampling period[23]. See Table S1 for further details on the samples.

**FIV detection and sequencing**. Total DNA was extracted from 50 μl whole blood samples using the QIAGEN DNeasy Blood & Tissue extraction kit (Qiagen, Inc., Valencia, CA) with an extended incubation period of two hours or from 200 μl whole blood samples using a phenol–chloroform extraction as per Pietro et al.[51]. Isolated DNA was quantified using a QuBit 2.0 fluorometer (Thermo-FisherScientific). DNA from individual puma were screened for the presence of $FIV_{pco}$ provirus using a specific qPCR assay as described by Lee et al.[52]. Using these data, we compared prevalence from both regions using a two-sample test for equality of proportions.

Full *ORFA* and *pol* gene regions were isolated from those samples identified as qPCR positive using a nested PCR protocol. See the Supplementary Note for the sequencing protocol and Table S8 for primer details. While we also sequenced the *env* gene, our assessment of the temporal signal (see next section) indicated many discrepancies in the data regarding the use of a molecular clock (strict or relaxed) to analyse these data. Resulting genetic sequences with chromatograms were checked, assembled, trimmed and aligned using the MUSCLE algorithm[53] using Geneious 7.0.6. Our *Pol* and *ORFA* datasets (GenBank accession: MN563193–MN563239) were compiled together with those sequences available in the public database Genbank, previously isolated from across the USA[54] (Fig. S1). Recombination was detected using RDP software V4[55]; parameters were set at default with linear topology. Events were determined as true if supported by three or more methods with $p$ values $<10e^{-3}$ combined with phylogenetic support. Recombination-free datasets were used for all downstream phylogenetic analyses.

**Viral phylogenetics**. To examine the broad placement of the $FIV_{pco}$ isolates sampled during this study, a maximum-likelihood tree was constructed for the $FIV_{pco}$ dataset comprised of all isolates recovered in the USA using PhyML[56] with

the TN93 + G + I model and aLRT branch support; branches with <80 support were collapsed using TreeGraph2[57].

We used TempEst[58] to assess data quality control of our generated data through root-to-tip regression and observed largely varying deviations from the regression line for many of the sequenced *env* genes. These deviations precluded the use of strict or relaxed molecular clock models to analyse the data, and we hence decided to move forward with the *pol* and *ORFA* sequence data. We used BEAST 1.10 [59] with BEAGLE 3.1[60] to perform both discrete and continuous phylogeographic analyses[61] based on the concatenated *pol* and *OFRFA* sequences from each subtype using an HKY substitution model (found most suitable for this smaller number of sequences using 'smart model selection' in PhyML[62]). For $FIV_{pco}$ CO, we performed Bayesian model selection on various model combinations comprising a strict molecular clock and an uncorrelated relaxed clock model with an underlying lognormal distribution, as well as three different coalescent models: a constant population size model, an exponential growth model and a non-parametric Bayesian skygrid model[63]. We also tested three relaxed random walk (RRW) models of continuous diffusion for the phylogeographic analyses and included each population (WUI and UB) as a discrete trait. All Bayesian model selection experiments were performed by (log) marginal likelihood estimation using path sampling and stepping-stone sampling[64–67] (see Table S9). Multiple replicates were run with different starting seeds to ensure convergence. Based on the outcome of the Bayesian model selection procedure, we presented the results obtained for the relaxed molecular clock models, the exponential population size coalescent model and the Cauchy RRW model. For $FIV_{pco}$ WY, as there were only 6 sequences, we applied a different approach as there was not enough data to obtain stable results for these complicated evolutionary models. In this case, we assumed a strict clock and constant population size and set the root prior to a uniform distribution (0, 300) reflecting our expectation that this subtype had been circulating for no more than 300 years. For both subtypes, duplicate MCMC chains were run for 200 million generations, with trees and parameters sampled every 20,000 steps. We used the program Tracer version 1.7[68] to examine ESS values (with parameter estimates accepted if the ESS was >200) and obtain HPD intervals for estimated parameters.

To identify the hidden population structure in our time-scaled phylogenies, we applied the 'tree structure' R package[69] using the default values. This analytical routine compares discrepancies between observed and idealised genealogies to identify clades under differing epidemiological or demographic processes[69]. As we did not have enough sequences to find meaningful structure in $FIV_{pco}$ WY, we limited this analysis to $FIV_{pco}$ CO. Any clades identified as significantly departing from the idealised geneology were further investigated using the R package 'phylodyn'[70] to estimate effective population size through time. This nonparametric Bayesian approach uses integrated nested Laplace approximation (INLA) to estimate effective population size efficiently, while accounting for preferential sampling by modelling the sampling times as a Poisson process (see ref. [71]).

**Host genomic data**. We genotyped 130 pumas (76 individuals from the UB and 54 individuals from the WUI) using a ddRADseq approach (see ref. [26] for sequence and bioinformatics details). From these genomic data, we quantified individual relatedness using the inverse proportion of shared alleles (Dps[72]) and used the resultant pairwise distance matrix in the downstream analyses. We also calculated the Smouse measure of individual genetic distance[39] to check if our results were sensitive to the distance measure used. We were unsuccessful in obtaining ddRAD data for seven individuals for which corresponding viral genomic data were available (see Fig. 1) and for these individuals we used the mean population relatedness value. Removing these individuals from the analyses did not qualitatively alter the results. Furthermore, we used the complete Dps dataset from all individuals and interpolated this distance across each landscape using a kriging approach. We converted this interpolated surface into a resistance raster in R and calculated resistance distances between each individual with $FIV_{pco}$ sequence data using a Circuitscape, which uses circuit theory to accommodate uncertainty in the route taken[73]. See https://github.com/nfj1380/ColoradoPumaFIVproject for details.

We estimated a coalescent-based phylogeny with bootstrap support values using SVD quartets[74] as implemented in PAUP[75]. SVD quartets, which is currently considered the most robust and computationally efficient SNP-based phylogenetic estimation method[37] and uses a site-based approach (considering each SNP has an independent genealogy) to estimate combinations of four-taxon relationships and heuristically summarise the resulting trees into a species tree phylogeny[37,74]. We evaluated a maximum of 100,000 random quartets using the QFM quartet assembly method and the multispecies coalescent tree model for generating the topology, and performed 100 bootstrap replicates for assessing topological branch support for all terminal taxa.

**Landscape data**. We collected Geographic Information Systems (GIS)-based landscape data that we hypothesized would be important for puma relatedness in Colorado (see Table S7 for more detail on GIS data sources and ecological justification for each landscape variable). This included percent impervious surface (e.g., roads, buildings, etc.), land cover (forested, open-natural, and human developed as sub-rasters), percentage tree canopy cover, vegetation density, rivers/

streams, roads, minimum temperature of the coldest month, annual precipitation, topographic roughness, and elevation. Resistance surfaces were created from this landscape data using the Reclassify and Raster Calculator tools in ArcGIS v. 10.1. We defined resistance/conductance cost values based on expert opinion as well the 'Resistance GA'[76] routine in R. Resistance GA optimises resistance/conductance surfaces based on the host pairwise genetic distances. We used this approach to test which landscape variables best approximated host gene flow and to formulate optimized resistance surfaces for use in viral spread analyses. To run the Resistance GA routine, we used an AIC objective function and tested all possible transformations (i.e. Ricker and Monomolecular transformations). As none of our landscape variables outperformed the null model in the UB region based on AICc, there was no need to further estimate the overall multivariate resistance surface shaping the host as more complex models get increasingly penalized in AICc scores (W. Peterman pers. comm). For the WUI region, canopy cover and urban landscape cover were within 2 AICc units of the best univariate model (spatial proximity), thus we included both features to create a multivariate optimised surface. We included resistance surfaces defined by expert opinion as well as host genetic optimized surfaces to help distinguish the effects of host genetics and landscape on $FIV_{pco}$ spread. Mantel tests were used to screen for collinearity between surfaces and for spatial autocorrelation prior to model construction.

We calculated the proportion of uninfected to infected individuals within a 5 km buffer (estimated average distance between individuals[77]) of each $FIV_{pco}$ positive individual using the 'summarize within' tool (also in ArcGIS v. 10.1). Estimates from each population were compared using a Mann–Whitney U Test.

### Statistics and reproducibility

*Impact of host relatedness and landscape resistance on viral spread.* We used generalized dissimilarity modelling (GDM)[35] to quantify if host relatedness and landscape shaped $FIV_{pco}$ spread for $FIV_{pco}$ CO. GDM is a flexible non-linear regression approach that fits monotonic I-spine functions to pairwise matrix data[35] to describe the rate and magnitude of, in this case, $FIV_{pco}$ phylogenetic change. We first calculated $FIV_{pco}$ patristic distance (using the maximum clade credibility tree) and converted this distance metric into a dissimilarity measure: $ds_{FIVple} = \left(\frac{d_i}{\max(d_{ij})}\right)^2$ where $d$ is pairwise distance. The host Dps matrix and all of the landscape resistance matrices were converted to dissimilarities the same way. Specifically, GDM uses generalized linear models (GLMs) to model $FIV_{pco}$ patristic distance in the form of:

$$-\ln(ds_x) = a_0 + \sum_{p=1}^{n} | f_p(x_{pi}) - f_p(x_{pj}) |$$

where $i$ and $j$ are individual puma, $a_0$ is the intercept, $p$ is the number of covariates and $f_p(x)$ have I-spline transformed versions of the predictors (see refs. [35,78] for further details). We used a backward elimination model selection approach and permutation tests ($n = 99$) to test for significance[35,79]. The model with the highest deviance (±2% deviance explained) with the smallest number of predictors was reported. We performed GDM using the same predictor sets as in Trumbo et al.[26] for each region but in our reduced dataset (i.e., with individuals with $FIV_{pco}$ data), temperature and elevation were strongly correlated with space (Mantel $r = 0.93$). We tested different values of $K$ (see below) and treated each as resistance or conductance surfaces yet this made no difference to the GDM models (we present results from resistance surfaces only with $K = 100$). Analysis of $FIV_{pco}$ WY was not possible given the small sample size.

We compared our non-linear GDM models to maximum likelihood of population-estimate (MLPE) models[36] to test the validity of our results. Instead of fitting non-linear splines, the MLPE approach uses linear mixed models to fit variables. We compared univariate MLPE models using Bayesian information criterion (BIC) model selection.

*Impact of environmental factors on viral dispersal velocity.* The analysis of the impact of environmental factors and host genetic differentiation on the dispersal velocity of viral lineages was performed using R functions of the package "seraphim"[80] (see refs. [81,82] for a similar workflow). In this analysis, each environmental factor, as well as the interpolated host genetic distance surface, was described by a raster that defines its spatial heterogeneity and that was used to compute an environmental distance for each branch in the phylogeny using two different path models: (i) the least-cost path model, which uses a least-cost algorithm to determine the route taken between the starting and ending points[83], and (ii) the Circuitscape path model. Here, we investigated the impact of the environmental rasters listed in Table S5 as well as the resistance raster generated from host genetic distance interpolation. We generated distinct land cover rasters from the original categorical land cover raster (resolution = 0.5 arcmin) by creating lower resolution rasters (2 arcmin) whose cell values equalled the number of occurrences of each land cover category within the 2 arcmin cells[81]. For each considered environmental factor, several distinct rasters were also generated by transforming original raster cell values with the following formula: $v_t = 1 + k*(v_o/v_{max})$, where $v_t$ and $v_o$ are the transformed and original raster cell values, and $v_{max}$ the maximum raster cell value recorded in the raster. The rescaling parameter $k$ here allowed the definition and testing of different strengths of raster cell conductance or resistance, relative to the conductance/resistance of a cell with a minimum value set to "1". For each environmental factor, we tested three different values for $k$ (i.e., 10, 100 and 1000). Finally, all these rasters were tested as potential conductance factors (i.e., factors facilitating movement) and as possible resistance factors (i.e., factors impeding movement). The statistic $Q$ was used to estimate the correlations between phylogenetic branch duration and environmental distances. $Q$ is defined as the difference between two coefficients of determination ($R^2$): (i) $R^2$ obtained when branch durations are regressed against environmental distances computed on the environmental raster, and (ii) $R^2$ obtained when branch durations are regressed against environmental distances computed on a null raster, i.e., an environmental raster with a value of "1" assigned to all the cells. For positive distributions of estimated $Q$ values (i.e., with at least 90% of positive values), statistical support was then evaluated against a null distribution generated by a randomization procedure and formalized as an approximated Bayes factor (BF) support [84]. To account for the uncertainty related to the Bayesian inference, this analysis was based on 1000 trees sampled from the post-burn-in posterior distribution inferred using the continuous phylogeographic model. We performed two distinct analyses, one per region, gathering all phylogenetic branches occurring on each study area.

*Ethics statement.* Puma samples were collected as part of ongoing studies by Colorado Parks and Wildlife (CPW) between 2006 and 2014. We handled all pumas in accordance with approved CPW Animal Care and Use Committee (ACUC) capture and handling protocols (ACUC file #08-2004, ACUC protocol #03-2007 ACUC 16-2008). Samples were provided to Colorado State University for diagnostic evaluation. Colorado State University and Colorado Parks and Wildlife (CPW) Institutional Animal Care and Use Committees reviewed and approved this work prior to initiation (CSU IACUC protocol 05-061A).

### Data availability

DNA sequences—GenBank accession numbers MN563193–MN563239. The sequence alignment file used to create the phylogenies in this paper as well as the data to reproduce the generalized dissimilarity models and phylogeographic models is available on GitHub: https://github.com/nfj1380/ColoradoPumaFIVproject.

### Code availability

All code is available on GitHub: https://github.com/nfj1380/ColoradoPumaFIVproject.

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

## Acknowledgements

This project was funded by a National Science Foundation Ecology of Infectious Diseases research program grants (DEB 1413925). M.C. was funded by the University of Minnesota's Office of the Vice President for Research and Academic Health Center Seed Grant. G.B. acknowledges support from the Interne Fondsen KU Leuven/Internal Funds KU Leuven under grant agreement C14/18/094, and the Research Foundation—Flanders ("Fonds voor Wetenschappelijk Onderzoek—Vlaanderen," G0E1420N). We thank Bill Peterman for his advice on the resistanceGA pipeline.

## Author contributions

N.F.J. conducted the analysis and wrote the initial draft of the paper to which all authors contributed. K.L. and M.A. studied the puma in the field and provided the blood samples. S.K., D.T., P.S., E.G. and S.V. collected virus and host genetic data. S.D., R.B. and G.B. contributed to the biogeographic and phylogenetic analyses. M.C., S.V., C.F., H.E. and S.C. conceived of the project.

## Competing interests

The authors declare no competing interests.
