## [Peer Review File · Communications Biology]

Reviewers' comments:

Reviewer #1 (Remarks to the Author):

Summary: This study compared 2 different study areas to understand the impact of urbanization on gene flow patterns, and resulting reduced contact and FIV spread in puma. Their study area with urbanization had landscape variables that predicted FIVpco spread, but not in the study area without urbanization.

I believe these studies are still at the novel stage with only a handful of similar type 'landscape disease genomics' studies published since the Biek first 2006 paper with of course, plenty of review papers pointing towards the need and importance of more landscape genetics into disease studies (2 of which this study cites and authors on). Here are two other references I did not see in your paper that I would suggest incorporating into your literature cited and intro or discussion sections.

Bruniche-Olsen A, Burridge CP, Austin JJ, Jones ME (2013) Disease induced changes in gene flow patterns among Tasmanian devil populations. *Biological Conservation* 165, 69-78.

Kyle CJ, Rico Y, Casillo S, Srithayakumar V, Cullingham C, White BN, Pond BA (2014) Spatial patterns of neutral and functional genetic variations reveal patterns of local adaptation in raccoon (*Procyon lotor*) populations exposed to raccoon rabies. *Molecular Ecology* 23, 2287-2298.

While I think the paper is excellent and the research question novel, I have some suggestions that I think might help strengthen overall findings and methodologies.

Abstract: (Puma concolor) here in abstract ?

Abstract: Ln 42 'whereas unpaved roads were more important in the unbounded region.' Isn't this misleading slightly? Suggest removing.

Ln 50-52: Not necessary to cite a ton of reviews, but there is also Schwabl et al 2016 on this subject

Schwabl P, Llewellyn M, Landguth EL, Andersson BA, Kitron U, Costales JA, Ocana S, Grijalva MJ (2016) Prediction and prevention of parasitic diseases using a landscape genomics framework. *Trends in Parasitology*. DOI: <http://dx.doi.org/10.1016/j.pt.2016.10.008>.

General: The choice of acronyms for study areas is slightly confusing, but this could be me, so I would ask others. When I see UB, I think urbanization and then I think your WUI is the wild one, so maybe think about a slight change there. Even, e.g., LN 91, you use 'urban-wildland interface' so maybe UWI would help. Unsure what to use for the unbounded one however, so perhaps it is a matter of reframing upfront: you are looking at 2 study areas with and without urbanization.

LN 114: Is this significant finding? Can you run a quick test and add to table?

Results: I am torn about showing Roads for UB.

Figure 3b: Are these partial effect plots? If so were all other variables held at median, mean?

Lns 400-403: Could you give more detail here? I looked through your code, but could not find any details on genetic resistance. How did you calculate resistance distances, for example?

Methods 434: Given the uncertainty still in LG methods results, I would recommend running 2 models to corroborate and strengthen your conclusions. While I see no issue with regression on distance matrices, MLPE is the preferred method with AIC as a model selection. See Shirk et al. 2018 comparison of methods for LG in *Molecular Ecology Resources*.

Likewise, while I personally see no issue with Dps, standard approaches with LG is to run a couple of measures to make sure results are robust on this side of the equation. PCA with more than 4 axes at least shows promise as well as Rousset's A.

Reviewer #2 (Remarks to the Author):

The study describes interesting and novel questions and it is based on a large sample size, which for a study of large mammals is not trivial. However, I find major flaws with the study design and the importance of the derived results. Therefore, I do not recommend publication of this manuscript in *Communications Biology*.

One of the main questions, also reflected in the title is how landscape connectivity influences spread of a virus through puma as a host and a transmission vector. However, the authors give very little attention to the landscape connectivity itself and the methods they use to define resistance surface (a base for any connectivity analysis) and connectivity modelling itself are described very vaguely, and based on this description and current state of art in landscape ecology they are simply incorrect. Authors did not use any empirical models to define the resistance surface. It seems that they applied a simple expert based model to define cost values of the resistance without explaining how exactly those values were assigned to each landscape component. Expert based resistance has proven to produce incorrect results (e.g. see Zeller et al. 2012). The method is highly criticized and this should not be accepted in the journal of such rank. Furthermore, it is surprising and disappointing that the authors having such a vast genetic dataset of puma population didn't make an effort to use empirical approach to correctly develop resistance surface.

Regarding the results, the authors derived several strong conclusions and made statements about the importance of the relationships between FIV spread and set of tested variables even though these relationships, except of one (spatial proximity), were statistically not significant (L190-194 and Fig.3). Furthermore, the reported variance explained by the WUI model of 20% is also not very high.

The study describes interesting and novel questions and it is based on a large sample size, which for a study of large mammals is not trivial. However, I find major flaws with the study design and the importance of the derived results. Therefore, I do not recommend publication of this manuscript in *Communications Biology*.

One of the main questions, also reflected in the title is how landscape connectivity influences spread of a virus through puma as a host and a transmission vector. However, the authors give very little attention to the landscape connectivity itself and the methods they use to define resistance surface (a base for any connectivity analysis) and connectivity modelling itself are described very vaguely and based on this description and current state of art in landscape ecology they are simply incorrect. Authors did not use any empirical models to define the resistance surface. It seems that they applied a simple expert-based model to define cost values of the resistance without explaining how exactly those values were assigned to each landscape component. Expert based resistance has proven to produce incorrect results (e.g. see Zeller et al. 2012). The method is highly criticized, and this should not be accepted in the journal of such rank. Furthermore, it is surprising and disappointing that the authors having such a vast genetic dataset of puma population didn't make an effort to use empirical approach to correctly develop resistance surface.

Regarding the results, the authors derived several strong conclusions and made statements about the importance of the relationships between FIV spread and set of tested variables even though these relationships, except of one (spatial proximity), were statistically not significant (L190-194 and Fig.3). Furthermore, the reported variance explained by the WUI model of 20% is also not very high.

Rebuttal

I apologize for the lengthy review. In addition to the holiday season, I had trouble with the 3rd reviewer, but we now decided to move forward as we have sufficient feedback.

Your manuscript entitled "Host relatedness and landscape connectivity shape pathogen spread in a large secretive carnivore" has now been seen by 2 referees. You will see from their comments below that while they find your work of great interest, some important points are raised. We are interested in the possibility of publishing your study in *Communications Biology*, but would like to consider your response to these concerns in the form of a revised manuscript before we make a final decision on publication.

We therefore invite you to revise and resubmit your manuscript, taking into account the points raised, specifically:

1. Do statistical test for Line 114 as suggested by Reviewer 1.

Response: We performed a 2-sample test for equality of proportions to quantify if prevalence differed by region and found that it does ($p = 0.04$). We have updated the results and methods accordingly (L116, L357-58).

2. Run 2 models to corroborate and strengthen findings as suggested by Reviewer 1 (methods 434) and to include more measures with the LG analysis as suggested by Reviewer 1.

Response: This is a great suggestion from Reviewer 1. Both approaches take different philosophies to fit the data. The generalized dissimilarity models (GDM) we presented in the previous version fit non-linear terms in an analogous way to GAMs, whereas methods such as maximum likelihood of population estimates (MLPE) fit linear mixed models. The non-independent nature of pairwise distance matrices is also accounted for differently: GDM uses permutations whereas MLPE treats pairs of taxa as a random effect. Nonetheless, the results from both approaches were similar. Most variables that explained deviance in our GDM models were also within 2 log units of each other in the MLPE model when we calculated BIC (Bayesian information criterion). We performed inference under the MLPE models and added the following text to the manuscript:

‘To support these results, we compared our non-linear GDM models to MLPE. One advantage of the GDM method over MLPE and other methods is that it can capture non-linear associations between response and predictor matrices. However, model performance of MLPE has been more rigorously evaluated compared to GDM on landscape genetic datasets³⁹. Here, we highlight factors that explain the most deviance in our GDM models and are within two log units of the best performing MLPE models (Tables S6 & S7). (L204-210).

We provide MLPE model results as supplementary tables (Tables S5/6) and have updated our methods as well. Non-linear models have been shown to have lower type I error than linear models in landscape genetics (Balkenhol et al, 2009) and this could be the case here, although these methods haven't been compared directly. We think this could be a useful future research direction.

Balkenhol, N., Waits, L.P. and Dezzani, R.J. (2009), Statistical approaches in landscape genetics: an evaluation of methods for linking landscape and genetic data. Ecography, 32: 818-830. doi:10.1111/j.1600-0587.2009.05807.x

3. Address the point on using empirical models to define resistance surface (Reviewer 2).

Response: Originally, we were aiming to make our paper comparable to Trumbo et al 2019 by using resistance surfaces configured the same way. However, we agree that using empirical models to define the resistance surfaces could be beneficial and have implemented these approaches in the revision. Please see detailed response to Reviewer 2 below.

4. Address the point about the statements on the importance of relationships b/t FIV spread and the variables, given that only one is significant. Authors may need to tone down the conclusions (Reviewer 2).

Response: We have toned down our conclusions in the discussion (L257-260, 294, 334) and reworded the results on the relationship between spread and our variables. As the MPLE results largely corroborated with our GDM results, we now have more support for our findings.

5. Address the point on the variance explained by the WUI model (Reviewer 2).

Response: Pumas can disperse long distances and disease transmission is a complex process, so explaining 0.2 deviance is higher than we anticipated. As Reviewer 1 points out, our study is novel and there are no direct comparisons. Other studies on large carnivores find similar model effect sizes (e.g. ~17% in Trumbo et al. 2019, ~25% in Curry et al. 2019, compared to 20% in this manuscript). Further, our WUI model fit was significant ($p = 0.028$) and our GDM results were further supported by our MLPE results.

Trumbo, D. et al. Urbanization impacts apex predator gene flow but not genetic diversity across an urban-rural divide. Mol. Ecol. 2020

Curry, Caitlin J et al. "Genetic analysis of African lions (Panthera leo) in Zambia support movement across anthropogenic and geographical barriers." PloS one vol. 14,5 e0217179.. (2019), doi:10.1371/journal.pone.0217179

6. Address all textual/figure/method clarifications by reviewers and addition of suggested references.

Response: We have addressed all textual/figure clarifications as well as adding the extra references as suggested.

Reviewers' comments:

Reviewer #1 (Remarks to the Author):

Summary: This study compared 2 different study areas to understand the impact of urbanization on gene flow patterns, and resulting reduced contact and FIV spread in puma. Their study area with urbanization had landscape variables that predicted FIVpco spread, but not in the study area without urbanization.

I believe these studies are still at the novel stage with only a handful of similar type 'landscape disease genomics' studies published since the Biek first 2006 paper with of course, plenty of review papers pointing towards the need and importance of more landscape genetics into disease studies (2 of which this study cites and authors on). Here are two other references I did not see in your paper that I would suggest incorporating into your literature cited and intro or discussion sections.

Bruniche-Olsen A, Burridge CP, Austin JJ, Jones ME (2013) Disease induced changes in gene flow patterns among Tasmanian devil populations. *Biological Conservation* 165, 69-78.
Kyle CJ, Rico Y, Casillo S, Srithayakumar V, Cullingham C, White BN, Pond BA (2014) Spatial patterns of neutral and functional genetic variations reveal patterns of local adaptation in raccoon (*Procyon lotor*) populations exposed to raccoon rabies. *Molecular Ecology* 23, 2287-2298.

Response: We agree that these references should be added and we now cite them in the introduction (L54).

While I think the paper is excellent and the research question novel, I have some suggestions that I think might help strengthen overall findings and methodologies.

Response: We thank the Reviewer for their positive and constructive comments.

Abstract: (Puma concolor) here in abstract ?

Response: Thank you. This has been added.

Abstract: Ln 42 'whereas unpaved roads were more important in the unbounded region.' Isn't this misleading slightly? Suggest removing.

Response: We have changed the abstract to the following to tone down the finding related to roads:

'The most important predictors of viral spread also differed; host spatial proximity, host relatedness, and mountain ranges played a role in FIV spread in the WUI, whereas roads might have facilitated viral spread in the unbounded region.' (L41-44).

Ln 50-52: Not necessary to cite a ton of reviews, but there is also Schwabl et al 2016 on this subject

Schwabl P, Llewellyn M, Landguth EL, Andersson BA, Kitron U, Costales JA, Ocana S, Grijalva MJ (2016) Prediction and prevention of parasitic diseases using a landscape genomics framework. *Trends in Parasitology*.

DOI: <http://dx.doi.org/10.1016/j.pt.2016.10.008>.

Response: Thank you – this is another relevant review on the topic to add. We have done so (L54).

General: The choice of acronyms for study areas is slightly confusing, but this could be me, so I would ask others. When I see UB, I think urbanization and then I think your WUI is the wild one, so maybe think about a slight change there. Even, e.g., LN 91, you use ‘urban-wildland interface’ so maybe UWI would help. Unsure what to use for the unbounded one however, so perhaps it is a matter of reframing upfront: you are looking at 2 study areas with and without urbanization.

Response: We agree that the acronyms are slightly confusing and there has been much debate amongst the authors on which terms to use. WUI is the term generally used to describe the wildland urban interface in landscape ecology (e.g. Radeloff et al 2005) and we think that changing this to UWI could lead to confusion too. We have now corrected the ‘urban-wildland interface’ phrasing to at least stay consistent with how we present “WUP”.

*Radeloff, V. C.; Hammer, R. B.; Stewart, S. I.; Fried, J. S.; Holcomb, S. S.; McKeefry, J. F. (2005). "The Wildland-Urban Interface in the United States". *Ecological Applications*. 15 (3): 799–805. doi:10.1890/04-1413.*

LN 114: Is this significant finding? Can you run a quick test and add to table?

Response: Good catch – we performed a 2-sample test for equality of proportions and found that this difference was significant. We have updated the results and methods accordingly (L116, L357-58).

Results: I am torn about showing Roads for UB.

Response: As roads explains a reasonable (yet insignificant) amount of deviance and roads are supported by some GPS movement data, we think retaining roads in Fig. 3 is appropriate. Also, roads are now amongst the most important predictors in our MLPE models for the UB. In light of this comment we have toned down the importance of roads in the abstract and in the discussion (L294).

Figure 3b: Are these partial effect plots? If so were all other variables held at median, mean?

Response: Yes, they are partial effect plots and all other variables were held at the mean. We have updated the figure caption to make this clear. The caption now reads: ‘The right panel shows partial effect plots (all other variables held at the mean).

Lns 400-403: Could you give more detail here? I looked through your code, but could not find any details on genetic resistance. How did you calculate resistance distances, for example?

Response: Thank you for this suggestion. We have updated the code to make it clear in which section we calculated genetic resistance distance (L60-150 of the code). We have

also added further text to clarify how we estimated landscape resistance. We have added the following:

'We defined resistance/conductance cost values based on expert opinion as well the 'Resistance GA'⁷⁴ routine in R. Resistance GA optimises resistance/conductance surfaces based on the host pairwise genetic distances. We used this approach to test which landscape variables best approximated host gene flow and to formulate optimized multivariate resistance surfaces for use in viral spread analyses.' (L454-59).

Methods 434: Given the uncertainty still in LG methods results, I would recommend running 2 models to corroborate and strengthen your conclusions. While I see no issue with regression on distance matrices, MLPE is the preferred method with AIC as a model selection. See Shirk et al. 2018 comparison of methods for LG in Molecular Ecology Resources.

Response: Running two models is a good suggestion. One advantage of the GDM method over MLPE and other methods is that it can capture non-linear associations between response predictor matrices, so comparing GDM output with MPLE output is a challenge. GDM also uses a permutation method for model selection rather than information theoretic approaches. However, we see the advantage of running MLPE as well to strengthen our results. We have added:

'To support these results, we compared our non-linear GDM models to MLPE. One advantage of the GDM method over MLPE and other methods is that it can capture non-linear associations between response and predictor matrices. However, model performance of MLPE has been more rigorously evaluated compared to GDM on landscape genetic datasets³⁹. Here, we highlight factors that explain the most deviance in our GDM models and are within two BIC log units of the best performing MLPE models (Tables S6 & S7). (L204-210).

Reassuringly, most of the factors that explained deviance in our GDM models were within 2 BIC units of the top-performing predictor. There were some differences though. The MLPE approach also found that host genetic resistance was important in the UB models, and roads and streams were important in the WUI models. We provide the full results in Tables S4-S5. Why our MLPE models were less conservative compared to GDM is unclear. Non-linear methods are also known to be less sensitive to type 1 error compared to linear methods (Balkenhol et al 2009), although these methods haven't been directly compared. As GDM is increasingly employed in landscape genetics since Fitzpatrick et al's 2014 paper, and whilst outside the scope of this paper, a direct comparison of these approaches is warranted and would be a valuable research direction.

Balkenhol, N., Waits, L.P. and Dezzani, R.J. (2009), Statistical approaches in landscape genetics: an evaluation of methods for linking landscape and genetic data. Ecography, 32: 818-830. doi:10.1111/j.1600-0587.2009.05807.x

Fitzpatrick, Matthew C.; Keller, Stephen R. (2014), Ecological genomics meets community-level modeling of biodiversity: mapping the genomic landscape of current and future environmental adaptation. Ecology Letters

Likewise, while I personally see no issue with Dps, standard approaches with LG is to run a couple of measures to make sure results are robust on this side of the equation. PCA with more than 4 axes at least shows promise as well as Rousset's A.

Response: This is a great suggestion. We also tested individual genetic distance calculated using the Smouse method (Smouse and Peakall, 1999) and found identical results. We have updated the methods and results accordingly (L217-218, L423-34).

Reviewer #2 (Remarks to the Author):

The study describes interesting and novel questions and it is based on a large sample size, which for a study of large mammals is not trivial.

Response: We thank the Reviewer for their constructive feedback. We wholly agree about the non-trivial nature of the sample size in this study.

However, I find major flaws with the study design and the importance of the derived results. Therefore, I do not recommend publication of this manuscript in *Communications Biology*. One of the main questions, also reflected in the title is how landscape connectivity influences spread of a virus through puma as a host and a transmission vector. However, the authors give very little attention to the landscape connectivity itself and the methods they use to define resistance surface (a base for any connectivity analysis) and connectivity modelling itself are described very vaguely, and based on this description and current state of art in landscape ecology they are simply incorrect. Authors did not use any empirical models to define the resistance surface. It seems that they applied a simple expert based model to define cost values of the resistance without explaining how exactly those values were assigned to each landscape component. Expert based resistance has proven to produce incorrect results (e.g. see Zeller et al. 2012). The method is highly criticized and this should not be accepted in the journal of such rank. Furthermore, it is surprising and disappointing that the authors having such a vast genetic dataset of puma population didn't make an effort to use empirical approach to correctly develop resistance surface.

Response: Thank you for stimulating this important discussion. Our methods were originally meant to be comparable to the most common approaches in landscape genetics, including work by our group on pumas in the same study area (Trumbo et al 2019 published in *Molecular Ecology*). These approaches relied on expert opinion to parameterize the resistance surface. In light of this comment, we see how empirical approaches could be more robust, so we have implemented the ResistanceGA approach to do so. This approach is very similar to the Zeller et al. 2012 approach, only instead of the pseudo-optimization approach used there, ResistanceGA tests all possible combinations with all possible variables. We were able to do this because we had relatively small geographic areas and large computational nodes on a high-performance computing cluster.

By clearly identifying geographic distance as the main predictor of host inter-individual genetic distance, our ResistanceGA results support isolation-by-distance as the main driver of host genetic differentiation (Table S3/S4). After discussion with William

Peterman (the author of the ResistanceGA paper), we decided there was no need to further estimate the overall multivariate resistance surface shaping the host.

Peterman, WE. ResistanceGA: An R package for the optimization of resistance surfaces using genetic algorithms. Methods Ecol Evol. 2018; 9: 1638– 1647. <https://doi.org/10.1111/2041-210X.12984>

Based on our initial screening and previous analysis, we identified a very low effect of spatial proximity on viral spread in the unbounded (UB) region (we detected a much stronger effect in the WUI). As there was a stronger effect of spatial proximity for the host in the UB, this indicated that there was a potential disconnect in this region between host and virus; subsequently we included all non-optimized landscape variables in our models. Our previous work in California (Lee et al. 2012 and Fountain-Jones et al. 2017) also shows a disconnect between host and pathogen landscape genetics. We have added a new paragraph to the results section to make the disconnect between host and virus clear to readers, and present our ResistanceGA findings (L185-198). In the methods we added: ‘We defined resistance/conductance cost values based on expert opinion as well the ‘Resistance GA’⁷⁵ routine in R. Resistance GA optimises resistance/conductance surfaces based on the host pairwise genetic distances. We used this approach to test which landscape variables best approximated host gene flow and to formulate optimized multivariate resistance surfaces for use in viral spread analyses.’ (L454-459)

Trumbo, D. et al. Urbanization impacts apex predator gene flow but not genetic diversity across an urban-rural divide. Mol. Ecol. (2020).

*Lee, J. S. et al. Gene flow and pathogen transmission among bobcats (*Lynx rufus*) in a fragmented urban landscape. Mol. Ecol. 21, 1617–1631 (2012).*

Fountain-Jones, N. M. et al. Urban landscapes can change virus gene flow and evolution in a fragmentation-sensitive carnivore. Mol. Ecol. 26, 6487–6498 (2017).

Regarding the results, the authors derived several strong conclusions and made statements about the importance of the relationships between FIV spread and set of tested variables even though these relationships, except of one (spatial proximity), were statistically not significant (L190-194 and Fig.3). Furthermore, the reported variance explained by the WUI model of 20% is also not very high.

Response: We agree and have toned down some of the statements in the results and discussion (L257-260, 294, 334).

As puma can disperse and potentially transmit FIV over long distances, explaining 20% deviance was actually higher than we expected. Further, our WUI GDM fit was significant ($p = 0.028$) and was further supported by the new MLPE results.

Reviewers' comments:

Reviewer #1 (Remarks to the Author):

Well done! The authors have addressed all of my comments quite thoroughly. I appreciate seeing the multimodel comparisons. I think this is a great manuscript and look forward to seeing it in print.

Reviewer #2 (Remarks to the Author):

The authors tried to address my main comment and concern regarding parametrization of the resistance surface. I appreciate their effort, the explanation provided in their response letter and also how they toned down the importance/significance of some of their results.

In this revised manuscript the authors tested an empirically-based approach using the ResistanceGA package and univariate models. However, I have concerns regarding implementation of this method in their study, for which, in general, they provided very scarce information in the manuscript. More information is needed on how exactly the optimization was done with ResistanceGA. Specifically, best practices in landscape genetics now is to optimize multivariately three things: which variables are in the model, their relative maximum resistance, and their functional form (sensu Shirk et al. 2010). There is not nearly enough information in the Methods to see how ResistanceGA was done to do this – e. g. It does not automatically vary both functional form and relative maximum resistance across multivariate space. In particular it is clear that the authors ONLY evaluated univariate response – which, even if properly optimized for RMax and functional form, is not an adequate measure of support for the null model compared to landscape resistance. Specifically, landscape resistance is a multi-scale, multivariate function – and it is very likely that a multivariate optimized model will greatly outperform any of the single variables. Therefore, the authors really should do a real and thorough multivariate multiscale optimization and compare those results with the null model. They say they consulted Peterman on this, but I would be very surprised that he would endorse concluding landscape resistance doesn't matter from incomplete and only univariate optimization without considering joint multivariate effects in optimized combination.

As far as I understand, at the end the authors built their resistance based on the expert based optimization (after testing empirical single variable optimization). However, if the empirical data from 217 individuals (!) do not allow resistance optimization, how can the authors assume that the expert opinion will do a better job? Zeller et al. (2017) showed high agreement between resistances optimized by genetic data and movement data for puma in southern California, thus perhaps authors should test different methods to optimize their resistance (?). In the worst case, it would be better to use the Zeller et al. 2017 resistance model, which is based on empirical data, although from different geographical area, than expert opinion.

References:

Shirk, A. J., Wallin, D. O., Cushman, S. A., Rice, C. G., & Warheit, K. I. (2010). Inferring landscape effects on gene flow: a new model selection framework. *Molecular ecology*, 19(17), 3603-3619.

Zeller, K. A., Vickers, T. W., Ernest, H. B., & Boyce, W. M. (2017). Multi-level, multi-scale resource selection functions and resistance surfaces for conservation planning: Pumas as a case study. *PLoS One*, 12(6).

Comments to the manuscript “Host relatedness and landscape connectivity shape pathogen spread in a large secretive carnivore”

The authors tried to address my main comment and concern regarding parametrization of the resistance surface. I appreciate their effort, the explanation provided in their response letter and also how they toned down the importance/significance of some of their results.

In this revised manuscript the authors tested an empirically-based approach using the ResistanceGA package and univariate models. However, I have concerns regarding implementation of this method in their study, for which, in general, they provided very sparse information in the manuscript. More information is needed on how exactly the optimization was done with Resistance GA. Specifically, best practices in landscape genetics now is to optimize multivariately three things: which variables are in the model, their relative maximum resistance, and their functional form (sensu Shirk et al. 2010). There is not nearly enough information in the Methods to see how ResistanceGA was done to do this – e. g. It does not automatically vary both functional form and relative maximum resistance across multivariate space. In particular it is clear that the authors ONLY evaluated univariate response – which, even if properly optimized for RMax and functional form, is not an adequate measure of support for the null model compared to landscape resistance. Specifically, landscape resistance is a multi-scale, multivariate function – and it is very likely that a multivariate optimized model will greatly outperform any of the single variables. Therefore, the authors really should do a real and thorough multivariate multiscale optimization and compare those results with the null model. They say they consulted Peterman on this, but I would be very surprised that he would endorse concluding landscape resistance doesn't matter from incomplete and only univariate optimization without considering joint multivariate effects in optimized combination.

As far as I understand, at the end the authors built their resistance based on the expert based optimization (after testing empirical single variable optimization). However, if the empirical data from 217 individuals (!) do not allow resistance optimization, how can the authors assume that the expert opinion will do a better job? Zeller et al. (2017) showed high agreement between resistances optimized by genetic data and movement data for puma in southern California, thus perhaps authors should test different methods to optimize their resistance (?). In the worst case, it would be better to use the Zeller et al. 2017 resistance model, which is based on empirical data, although from different geographical area, than expert opinion.

References:

Shirk, A. J., Wallin, D. O., Cushman, S. A., Rice, C. G., & Warheit, K. I. (2010). Inferring landscape effects on gene flow: a new model selection framework. *Molecular ecology*, 19(17), 3603-3619.

Zeller, K. A., Vickers, T. W., Ernest, H. B., & Boyce, W. M. (2017). Multi-level, multi-scale resource selection functions and resistance surfaces for conservation planning: Pumas as a case study. *PLoS One*, 12(6).

Responses are in bold below.

Reviewers' comments:

Reviewer #1 (Remarks to the Author):

Well done! The authors have addressed all of my comments quite thoroughly. I appreciate seeing the multimodel comparisons. I think this is a great manuscript and look forward to seeing it in print.

Response: Thank you! We thank the reviewer for the feedback.

Reviewer #2 (Remarks to the Author):

The authors tried to address my main comment and concern regarding parametrization of the resistance surface. I appreciate their effort, the explanation provided in their response letter and also how they toned down the importance/significance of some of their results.

In this revised manuscript the authors tested an empirically-based approach using the ResistanceGA package and univariate models. However, I have concerns regarding implementation of this method in their study, for which, in general, they provided very sparse information in the manuscript. More information is needed on how exactly the optimization was done with Resistance GA. Specifically, best practices in landscape genetics now is to optimize multivariately three things: which variables are in the model, their relative maximum resistance, and their functional form (sensu Shirk et al. 2010). There is not nearly enough information in the Methods to see how ResistanceGA was done to do this – e. g. It does not automatically vary both functional form and relative maximum resistance across multivariate space. In particular it is clear that the authors ONLY evaluated univariate response – which, even if properly optimized for RMax and functional form, is not an adequate measure of support for the null model compared to landscape resistance. Specifically, landscape resistance is a multi-scale, multivariate function – and it is very likely that a multivariate optimized model will greatly outperform any of the single variables. Therefore, the authors really should do a real and thorough multivariate multiscale optimization and compare those results with the null model. They say they consulted Peterman on this, but I would be very surprised that he would endorse concluding landscape resistance doesn't matter from incomplete and only univariate optimization without considering joint multivariate effects in optimized combination.

Response: Thanks for pointing out that our decision making regarding the resistance surface construction was unclear. After further discussions with Dr Peterman (a world leader in landscape resistance modelling, whom we have now thanked in the Acknowledgements section and cited as “personal communication” in the Landscape Data Methods), we determined a multivariate surface may outperform the null model,

but only for the WUI (wildland urban interface) landscape. For the unbounded landscape (UB), our results from the univariate surface (Table S4) suggest that there is no spatial signal in this data, as the ‘spatial proximity’ surface > 2 AIC’s of the null surface with no landscape variables included. Spatial proximity is statistically equivalent to an Intercept only model and further multivariate optimization should not improve the model (as the increase in parameters is penalized by AICc scores). The univariate results were less clear cut for the WUI (Table S5), but as Reviewer 2 suggested, we did construct a host genetic optimized multivariate surface. As to test every single possible multivariate surface would be too computationally demanding and outside the scope of this paper, we constructed a surface using the two variables within 2 AICc units from spatial proximity in our univariate results (“canopy cover” and “urban land cover”, Table S\5).

We added this optimized surface to our FIV spread model and interestingly the resulting Circuitscape pairwise matrix was strongly correlated with our host genetic resistance matrix (Mantel $\rho = 0.76$, $p = 0.001$) generated using our interpolation approach. Substituting either surface did not change behaviour of our FIV spread model (e.g., nearly identical amounts of deviance explained, marginal increase in significance ($p = 0.01$)), therefore we present the results with the optimized multivariate surface in the main text (Fig. 3) and the analysis with host genetic resistance in the supplementary materials (the new Fig. S5).

In the Results section, our manuscript now reads:

‘In the UB, our optimization approach revealed that none of the landscape variables (Table S3) explained host gene flow more than the null model (i.e. were > 2 AICc units higher than the null or the model with no landscape variables included, Table S4). Subsequently, no host-genetic optimized surfaces were included in the UB model (see Methods). In contrast, our optimisation approach identified a stronger impact of spatial proximity on host gene flow in the WUI (Table S5). Canopy cover and urban land cover univariate models were also within 2 AICc units of the spatial proximity model, thus we combined these variables together to generate a multivariate, host genetics optimised resistance surface (hereafter called host-optimized resistance surface). The host-optimized resistance surface was strongly correlated with interpolated host genetic resistance (Mantel $\rho = 0.76$, $p = 0.001$). We thus present models in the WUI region with host genetic resistance and host-optimised resistance included separately. In both regions, to tease apart the effect of host genetics and landscape on FIV spread, we also included the non- host genetics optimized landscape resistance/conductance surfaces (hereafter called landscape variables) in our models as well as spatial proximity and host variables.’ (L191-205)

We have also updated the methods to reflect these additions (L 463-71).

We have also added further information about how we performed the Resistance GA routine. We have added the following text:

... ‘To run the Resistance GA routine, we used an AIC objective function and tested all possible transformations (Ricker and Monomolecular transformations)’ (L462-63)

Further, we have updated the GitHub repository with the updated code and data that allows readers to fully replicate our analyses.

As far as I understand, at the end the authors built their resistance based on the expert based optimization (after testing empirical single variable optimization). However, if the empirical data from 217 individuals (!) do not allow resistance optimization, how can the authors assume that the expert opinion will do a better job? Zeller et al. (2017) showed high agreement between resistances optimized by genetic data and movement data for puma in southern California, thus perhaps authors should test different methods to optimize their resistance (?). In the worst case, it would be better to use the Zeller et al. 2017 resistance model, which is based on empirical data, although from different geographical area, than expert opinion.

Response: Thank you again for stimulating another important discussion in our group. A key part of this project is to separate the effects of landscape and host genetics in shaping FIV spread. Your suggestion to include multivariate host-genetics optimised surfaces was very useful as it further confirmed that in the WUI region spatial proximity coupled with host genetics was important in shaping the pattern of FIV spread. Host genetics (dPS) and host spatial proximity each explained over double the deviance (7% to ~20%) compared to the multivariate optimized surface in our model (spatial proximity was also significant in the model, but the all of other variables included were not). We think keeping both optimised and expert opinion-based surfaces in the GDM model is advantageous as it avoids any potential circularity (having landscape predictors that were not calibrated with host genetics, as host genetics is a predictor in the model). Including an optimised surface as well includes a variable that captures our best estimate of how landscape variables most align to host geneflow.

Taken together, we feel this additional analysis and detail adds extra rigour and further strengthens the key message in our paper on the importance of puma genetics and space in puma in shaping FIV spread at the urban edge.

References:

Shirk, A. J., Wallin, D. O., Cushman, S. A., Rice, C. G., & Warheit, K. I. (2010). Inferring landscape effects on gene flow: a new model selection framework. *Molecular ecology*, 19(17), 3603-3619.

Zeller, K. A., Vickers, T. W., Ernest, H. B., & Boyce, W. M. (2017). Multi-level, multi-scale resource selection functions and resistance surfaces for conservation planning: Pumas as a case study. *PLoS One*, 12(6).

REVIEWERS' COMMENTS:

Reviewer #2 (Remarks to the Author):

Thank you for addressing all my comments and rethinking some of the approaches to model connectivity.

I think that the authors have done a great job in improving the paper and I'm happy with this version to be published without further changes.